# Oral DhHP-6 for the Treatment of Type 2 Diabetes Mellitus

**DOI:** 10.3390/ijms20061517

**Published:** 2019-03-26

**Authors:** Kai Wang, Yu Su, Yuting Liang, Yanhui Song, Liping Wang

**Affiliations:** 1School of life Sciences, Jilin University, Changchun 130012, China; wangkai17@mails.jlu.edu.cn (K.W.); suyu17@mails.jlu.edu.cn (Y.S.); liangyt17@mails.jlu.edu.cn (Y.L.); yhsong16@mails.jlu.edu.cn (Y.S.); 2Key Laboratory for Molecular Enzymology and Engineering, the Ministry of Education, Jilin University, Changchun 130012, China

**Keywords:** type 2 diabetes mellitus, Deuterohemin-βAla-His-Thr-Val-Glu-Lys, high stability, anti-diabetic drug

## Abstract

Type 2 diabetes mellitus (T2DM) is associated with pancreatic β-cell dysfunction which can be induced by oxidative stress. Deuterohemin-βAla-His-Thr-Val-Glu-Lys (DhHP-6) is a microperoxidase mimetic that can scavenge reactive oxygen species (ROS) in vivo. In our previous studies, we demonstrated an increased stability of linear peptides upon their covalent attachment to porphyrins. In this study, we assessed the utility of DhHP-6 as an oral anti-diabetic drug in vitro and in vivo. DhHP-6 showed high resistance to proteolytic degradation in vitro and in vivo. The degraded DhHP-6 product in gastrointestinal (GI) fluid retained the enzymatic activity of DhHP-6, but displayed a higher permeability coefficient. DhHP-6 protected against the cell damage induced by H_2_O_2_ and promoted insulin secretion in INS-1 cells. In the T2DM model, DhHP-6 reduced blood glucose levels and facilitated the recovery of blood lipid disorders. DhHP-6 also mitigated both insulin resistance and glucose tolerance. Most importantly, DhHP-6 promoted the recovery of damaged pancreas islets. These findings suggest that DhHP-6 in physiological environments has high stability against enzymatic degradation and maintains enzymatic activity. As DhHP-6 lowered the fasting blood glucose levels of T2DM mice, it thus represents a promising candidate for oral administration and clinical therapy.

## 1. Introduction

Type 2 diabetes mellitus (T2DM) has increased in prevalence and emerged as one of the largest public health problems globally [1]. Type 2 diabetes mellitus (T2DM) has increased in prevalence and emerged as one of the largest public health problems globally. According to the World Health Organization (WHO) report on diabetes, the number of people suffering from diabetes rose from 108 million to 422 million from 1980 to 2014, respectively [2]. The global prevalence of diabetes in adults according to the International Diabetes Federation (IDF) report published in 2013 was 8.3%, and the global prevalence is expected to rise to 10.1% [3]. T2DM is considered as one of the fifth leading causes of death in the world. T2DM is characterized by obesity and insulin resistance due to an inadequacy of pancreatic β-cell mass and function [3,4]. Excessive oxidative stress contributes to the pathogenesis of T2DM by promoting β-cell dysfunction and reducing the sensitivity of peripheral tissues to insulin [5]. Oxidative stress induced by excessive reactive oxygen species (ROS) has been linked to a range of pathologies [6]. ROS generated in aerobic organs and tissues comprises free radicals, lipids, proteins, and nucleic acids [7]. ROS can penetrate the cell membrane and destroy the β-cells of the pancreas. Oxidative stress modifies many signaling pathways within a cell that can ultimately lead to insulin resistance [8]. Oxidative stress can reduce insulin sensitivity and damage insulin-producing cells in the pancreas [5]. The production of ROS in diabetes impairs the phosphatidylinositol 3-kinase (PI3K)/AKT pathway. The PI3K/AKT pathway plays an important role in the insulin signaling pathway, which regulates gluconeogenesis and glycogen synthesis [9,10]. Microperoxidases generated by the covalent binding of digested cytochrome c to oligopeptides are iron-porphyrin mini-enzymes that can eliminate ROS [11,12]. The supplementation of exogenous microperoxidases represents a new direction for T2DM studies.

DhHP-6 (Deuterohemin-βAla-His-Thr-Val-Glu-Lys) was designed based on the microperoxidase-11 (MP-11), generated through cytochrome c mediated hydrolysis [13]. The deuterohemin group has antioxidant properties, whereas the linear peptide AHTVEK improves water solubility and is highly conserved in MP and APx [13,14]. In our previous studies, DhHP-6 extended the lifespan of *Caenorhabditis elegans* (*C. elegans*) through its antioxidant activity [15,16]. DhHP-6 also ameliorates Alzheimer’s disease-related pathology and cognitive decline in APPswe/PSEN1dE9 transgenic mouse models [17]. When injected, DhHP-3 (deuterohemin-Ala-His-Lys) improves the symptoms of T2DM rats [18].

Peptides and protein drugs have gained popularity for the treatment of numerous diseases, including T2DM, due to their high selectivity and potent activity [19]. Protein and peptide drugs are commonly administered by injection because of low bioavailability by other routes [20]. Injections often cause pain, trauma, and distress to patients, resulting in poor patient compliance [21]. Although oral administration remains the most convenient route for clinical therapy, it is impracticable for protein and peptide drugs. Enzymatic degradation in low pH conditions and poor penetration of the intestinal membrane lead to low oral bioavailability [22,23]. To circumvent this, several strategies including nanoparticles, intestinal patches, structural modifications, microparticles, hydrogels, and capsules have been employed as oral delivery systems, with varying levels of success [20,24,25].

Cationic porphyrins can interact with lipid bilayers and reduce their stability [26]. Quadraplexes can be stabilized by porphyrins with a high number of charges [27]. Porphyrins can form stable complexes with bovine serum albumin (BSA) [28], and compounds with cationic porphyrins, such as DhHP-6, can become more stable and penetrate membrane bilayers more easily.

In order to verify the possibility of DhHP-6 as an oral drug for the treatment of T2DM, we examined the stability of DhHP-6 in simulated gastrointestinal (GI) fluid and plasma, and found that DhHP-6 had high stability. In addition, we examined the transmembrane transport capacity of DhHP-6 in Caco-2 cells. The Papp of DhHP-6 treated by simulated gastric fluid (SGF) or simulated intestinal fluid (SIF) was found to be greater than 10^−6^ cm/s, meaning that DhHP-6 was effectively absorbed in the body and verified in rats. DhHP-6 can effectively improve oxidative damage induced by H_2_O_2_ and improve insulin secretion in INS-1 cells. We also examined the effect of oral DhHP-6 in a T2DM mouse model, and found that DhHP-6 could effectively reduce the symptoms of blood glucose and insulin resistance. All these results showed that DhHP-6 could be used as an oral drug for the treatment of type 2 diabetes.

## 2. Results

### 2.1. Stability of DhHP-6 in Simulated Gastrointestinal Fluid

Protein and peptide drugs are degraded in the digestive tract due to the activity of gastrointestinal (GI) resident enzymes including pepsin and trypsin. To predict the stability of DhHP-6 in the GI tract, its stability was assessed in simulated GI fluid. As shown in Figure 1A, DhHP-6 was rapidly degraded (67.18% degraded within 30 min), which then slowed with 12.70% degradation in the subsequent 90 min. To study the degradation products of DhHP-6 in SGF, the final reaction was analyzed by HPLC-MS. Although the majority of DhHP-6 was degraded to DhHP-5 (Deuterohemin-βAla-His-Thr-Val-Glu, molecular weight: 1101.40 Da) (Figure 1B), the relative enzyme activity of the degradation products improved (Figure 1C). Interestingly, DhHP-6 was slowly degraded in SIF with approximately 79.74% remaining at the end of the reaction (Figure 2A). The degradation products were DhHP-5 and DhHP-2 (Deuterohemin-βAla-His, molecular weight: 772.64 Da) (Figure 2B). The relative enzymatic activity of the degradation products had no difference to that of DhHP-6 (Figure 2C).

### 2.2. Stability of DhHP-6 in Human Plasma

Peptide drugs absorbed in the blood are more sensitive to proteases, reducing their bioavailability. To explore its stability, DhHP-6 was incubated in human plasma and the samples collected at scheduled intervals analyzed by HPLC. Approximately 20.24% DhHP-6 was degraded over one day, and 60.63% remained after seven days (Figure 3A). The major degradation product was DhHP-3 (Deuterohemin-βAla-His-Thr) (Figure 3B) which retained the relative enzyme activity of DhHP-6 (Figure 3C).

### 2.3. Permeability of Digestion Products of DhHP-6 in Caco-2 Cells

The permeability of the DhHP-6 digestion products was determined in Caco-2 cells. Generally, drugs with a Papp ≥10^−6^ cm/s can be easily absorbed in vivo. The Papp values of the digestion products of DhHP-6 treated with SGF or SIF were greater than 10^−6^ cm/s (Table 1).

### 2.4. In Vivo Oral Absorption of DhHP-6

DhHP-6 was administered by intragastric (IG) administration or intravenous (IV) injection and blood samples were assessed by HPLC. The concentration of DhHP-6 in the blood was maximal at 15 min post-administration in both groups, which sharply reduced over the next 15 min (Figure 4). The concentration of DhHP-6 in the blood following IV injection was 1-fold higher than IG administration. The degradation of DhHP-6 using both administration methods was reduced, in accordance with its stability in plasma. DhHP-6 was again primarily degraded into DhHP-3.

### 2.5. DhHP-6 Ameliorates the Function and Mass of INS-1 Cells

The dysfunction of pancreas islets is highlighted by a loss of pancreatic β-cell function and mass. To determine the effects of DhHP-6 on pancreas islets, we assessed the proliferation and function of INS-1 cells. DhHP-6 promoted INS-1 proliferation in a concentration-dependent manner (Figure 5A). Notably, DhHP-6 (32 μM) led to a 0.37-fold increase in cell proliferation. DhHP-6 also suppressed H_2_O_2_-mediated damage in a concentration-dependent manner (Figure 5B). DhHP-6 promoted significant insulin secretion in INS-1 cells (Figure 5C) but did not stimulate lactic dehydrogenase LDH release. DhHP-6 therefore promoted the growth and insulin secreting ability of INS-1 cells (Figure 5D).

### 2.6. DhHP-6 Mitigates the Symptoms of T2DM Mice

DM causes high blood glucose and weight loss. To assess the effects of DhHP-6 on body weight and blood glucose, its effects were analyzed in T2DM mice. As shown in Figure 6A, the body weights of T2DM mice were higher than the control group at the beginning of the experiment, which decreased over the next four weeks. There were no significant improvements in the DhHP-6- or metformin-treated groups. However, the levels of fasting blood glucose of both DhHP-6- and metformin-treated groups were reduced compared to those of T2DM mice (Figure 6B).

Glucose tolerance was determined through oral glucose tolerance test (OGTT) assays. The blood glucose of T2DM mice was significantly elevated compared to that of the control group, which declined following DhHP-6 or metformin treatment (Figure 6C). In insulin tolerance test (ITT) assays to evaluate glucose clearance, T2DM mice were less sensitive to insulin than the control group, whereas both DhHP-6 and metformin improved insulin sensitivity (Figure 6D).

### 2.7. DhHP-6 Restores the Structure and Function of Pancreas Islets in T2DM Mice

As shown in Figure 7A, STZ injection in T2DM mice resulted in severe islet damage, leading to atrophy and cell vacuolar degeneration. Although vacuolar degeneration remained severe in mice treated with 20 mg/kg DhHP-6, the structure of the islets was ameliorated. Notably, the islets and vacuolar degeneration improved in both metformin- and 60 mg/kg DhHP-6-treated groups. Upon determination of the serum insulin levels by ELISA, serum insulin was low in T2DM mice, but increased after treatment with metformin and DhHP-6 (Figure 7B).

### 2.8. DhHP-6 Ameliorates Dyslipidemia in T2DM Mice

Blood lipid disorders (HDL-C, LDL-C, cholesterol, and triglycerides) often lead to insulin resistance [2]. Compared to the control group, the serum levels of HDL-C were significantly lower in T2DM mice, which were restored by DhHP-6 at concentrations of 20 and 60 mg/kg bodyweight (Figure 8A). The levels of LDL-C, cholesterol, and triglycerides in T2DM mice increased more significantly than the control group, and symptoms markedly improved following DhHP-6 or metformin treatment (Figure 8B–D).

### 2.9. DhHP-6 Improves Liver Damage in T2DM Mice

The liver regulates glycolipid metabolism. The ability of DhHP-6 to ameliorate dyslipidemia in T2DM mice was investigated through hepatic histological assessments. As shown in Figure 9A,B, compared to the control group, hepatic lobules were disorganized and liver cells in T2DM mice were irregular and swollen and showed signs of fat degeneration including fat vacuoles in the cytoplasm. In the group treated with 20 mg/kg DhHP-6, although the structure of liver tissue showed no significant improvement, the fat vacuoles were less severe than untreated groups. In addition, in metformin- and 60 mg/kg DhHP-6-treated groups, lower levels of steatosis were evident and the liver cells became more regular than T2DM mice. To further examine liver function, alanine transaminase (ALT), aspartate transaminase (AST), and alkaline phosphatase (ALP) were measured in serum samples. As shown in Figure 9C–E, although all classical markers of T2DM increased, both the metformin- and DhHP-6-treated groups led to a significant reduction in ALT, AST, and ALP compared to the model group, consistent with the results obtained from liver sections.

## 3. Discussion

DhHP-6 is a MP-11 mimetic with a porphyrin ring at its N-terminus. DhHP-3, a homolog of DhHP-6, mitigated the symptom of rats with T2DM by intraperitoneal injection [18]. In our previous studies, we found that linear peptides covalently attached to the porphyrin ring became more stable in plasma. In this study, the effects of DhHP-6 as an oral anti-diabetic drug were investigated.

The stability of the lipid bilayer can be decreased by interaction with cationic porphyrins [26] which can be beneficial to drug absorption. Porphyrin-BSA complexes also have higher stability [28]. In this study, DhHP-6 with a cationic porphyrin ring was more stable in SGF, SIF, and plasma. Both DhHP-6 and its degradation products displayed high permeability in Caco-2 cells and could be absorbed in the blood following IG administration. Those results meant that DhHP-6 was stable in vitro and in vivo. In addition, although DhHP-6 was degraded into DhHP-5 in SGF, DhHP-2 in SIF, or DhHP-3 in vivo, the relative enzymatic activity remained almost identical to that of DhHP-6. Although DhHP-6 changed in vitro and in vivo, the activity of the degraded products remained, which meant DhHP-6 could act as an oral drug.

T2DM is a chronic metabolic disorder syndrome characterized by hyperglycemia and dyslipidemia [29]. T2DM afflicts about 383 million adults, emerging as a serious health burden [30,31]. Oxidative stress is closely associated with the pathogenesis of T2DM. Increased oxidative stress is induced by several abnormalities that lead to a decline in endogenous antioxidant defenses, including hyperglycemia, insulin resistance, and lipid disorders [32,33]. Several strategies have been used to reduce oxidative stress through the supply of antioxidants (e.g., superoxide dismutase, catalase, vitamins A, C, E, folate, and trace elements) [13]. DhHP-6 extended the lifespan of *C. elegans* through its ability to enhance antioxidant capacity by scavenging ROS [15,16]. Injection of DhHP-3 improved the symptoms of T2DM rats. In this study, the degradation products of DhHP-6 in vitro had the same relative enzymatic activity as DhHP-6. DhHP-6 improved insulin secretion and cell damage induced by H_2_O_2_ in INS-1 cells. Excessive ROS can induce oxidative damage to organs associated with metabolic process, like the liver and kidney. Moreover, T2DM can lead to liver dysfunction, such as fatty liver, liver fibrosis, and non-alcoholic fatty liver disease (NAFLD) [2,34,35]. In line with these studies, hepatic lobules were obsolete and the permutation of liver cells caused them to be irregular and swollen in T2DM mice. After treatment with 60 mg/kg DhHP-6, steatosis was obviously alleviated, liver cells became more regular than those in T2DM mice. ALT, AST, and ALP in serum are considered markers of liver function [30]. ALT, AST, and ALP leaked from hepatocyte cytoplasm into the serum are considered an outcome of hepatic injury [2]. In this study, enzymic markers of liver function in serum increased in T2DM mice and decreased after treatment with DhHP-6, which was in accordance with the liver tissue histochemistry.

T2DM is characterized by hyperglycemia, dyslipidemia, and insulin resistance, which precedes insulin deficiency due to β-cell failure [30,36]. In the present study, the levels of fasting blood glucose in T2DM mice treated with DhHP-6 decreased significantly compared with those in T2DM mice. Insulin resistance (IR) and β-cell dysfunction lead to insulin deficiency, a hallmark of T2DM [37]. IR refers to a complex pathological process of insensitive response to the insulin hormone in insulin-targeted cells [38]. IR disables insulin-dependent tissues and cells to take up and utilize glucose. Thus, improving insulin resistance is beneficial to the improvement of T2DM symptoms. In the present study, we found that DhHP-6 ameliorated glucose tolerance and insulin resistance using OGTT and ITT and improved blood lipid disorders, demonstrating its ability to decrease insulin resistance in T2DM models. Hyperglycemia is closely related to the formation of reactive oxygen species (ROS) and weakens antioxidant defense, enhancing oxidative stress [39]. Oxidative stress can cause the loss of structure and function of pancreas islets. In this study, DhHP-6 treatment improved the levels of serum insulin in T2DM mice. In addition, islet structural damage induced by T2DM was restored after treatment with DhHP-6. Therefore, DhHP-6 represents a promising candidate for oral administration.

## 4. Materials and Methods

### 4.1. Materials

STZ used to establish the diabetic rat model was purchased from Sigma-Aldrich (Shanghai, China). Pepsin, trypsin, and Dulbecco’s Modified Eagle’s Medium (DMEM) for Caco-2 cell culture were purchased from Gibco (Shanghai, China). RPMI-1640 for INS-1 cell culture, penicillin, and streptomycin were purchased from Hyclone (Waltham, MA, USA). Fetal bovine serum (FBS) was purchased from Biological Industries (Kibbutz Beit-Haemek, Israel). Human plasma for stability detection was obtained from the People’s Liberation Army Hospital No. 461 (Changchun, China). Rat insulin ELISA kits were purchased from Mercodia (Guangzhou, China).

### 4.2. Stability in Simulated Gastrointestinal Fluid

Simulated intestinal fluid (SIF) and simulated gastric fluid (SGF) were prepared in accordance with the US Pharmacopeia proposed by the United States Pharmacopeial Convention in 2006 [40,41,42]. The pH of SGF was adjusted to 1.2 and that of SIF to 6.8 ± 0.1, which are similar to the mean pH in the rat stomach and intestine. DhHP-6 was dissolved in 5 mL SGF to a final concentration of 1 mg/mL at 37 °C shaking at 110 rpm for 0, 15, 30, 45, 60, 90, and 120 min, respectively. Samples (100 μL) were removed from the reaction and those from SGF were mixed with 30 μL 200 mmol/L Na_2_CO_3_ solution. Samples from SIF were heated to 95 °C to terminate the reaction. The final reaction solution (20 μL) was determined by HPLC-MS.

### 4.3. Peroxidase Activity of the Digestion Products

As a peroxidase mimetic, the enzymatic potency of the digestion products of DhHP-6 were assessed using the H_2_O_2_–Vitamin C (Vc) double-substrate method. Digestion products of DhHP-6 were diluted to 10 μM. Samples containing no enzyme were used as a control. The reaction system consisted of phosphate buffer solution (PBS) (50 mM, pH 7.4), H_2_O_2_ (1 mM), Vc (250 mM), and 10 μM DhHP-6. All samples were assayed at 25 °C. The decrease in the absorbance at 290 nm was measured for 30 s by UV–Vis spectrophotometer (Shimadzu, Kyoto, Japan), and measurements were performed prior to the addition of the digestion products and set as 100% relative enzymatic activity.

### 4.4. Stability of DhHP-6 in Plasma

DhHP-6 (dissolved in PBS) was incubated with human plasma in a thermostat water bath at 37 °C. An equal volume of PBS was incubated with human plasma as a control. Samples were collected at days 1, 2, 3, 4, and 6, and other proteins were removed through the addition of two volumes of acetonitrile. Samples after centrifugation were analyzed by HPLC.

### 4.5. Permeability of Digestion Products in Caco-2 Cells

Caco-2 cells were cultured in DMEM containing 10% FBS, 2 mM L-glutamine, 1% penicillin–streptomycin, and 1% non-essential amino acids. Cells were cultured in 25 mL culture flasks and incubated at 37 °C in 5% CO_2_ atmosphere (90% relative humidity). Approximately 250,000 cells/cm^2^ were seeded onto polycarbonate membranes of the inserts in 24-transwell plates in 600 mL of culture medium. The culture medium was changed every other day in week 1, and daily up until the end of the third week. Plates meeting a transepithelial electrical resistance >600 Ω/cm^2^ were used for drug transport experiments [43,44,45,46]. The culture medium of both sides of the monolayer was discarded, cells were washed three times in serum-free medium, and incubated for 30 min at 37 °C. Cells were replaced with 500 μL of fresh buffer solution on the basolateral side and with 500 μL sample solution (prepared in PBS and filtered through 0.22 μm sterile membranes) on the apical side. Samples (200 μL) were collected from the basolateral side after incubation at 37 °C for 120 min for HPLC analysis.

### 4.6. In Vivo Oral Absorption of DhHP-6

A total of 10 male Wistar rats (weight, 200 ± 20 g, provided by the Model Animal Research Center of Nanjing University in China) were randomly divided into two groups. One group was injected with 1 mg/mL DhHP-6 diluted in physiological saline and the other group was orally administered the same volume of DhHP-6. Blood samples were collected by retro-bulbar sinus puncture at scheduled intervals. Samples at 0 h were collected before DhHP-6 administration. The animal experimental procedure was conformed to the AnimalEthical Standards and Use Committee at Jilin University (The Animal Care Committee of Jilin University (License No.: 20160518)).

### 4.7. INS-1 Cell Culture

INS-1 cells were cultured in RPMI 1640 containing 10% FBS, 50 μM β-mercaptoethanol, 1% penicillin–streptomycin, 2 mM L-glutamine, and 1 mM sodium pyruvate at 37 °C in a humidified atmosphere (5% CO_2_) [47].

### 4.8. Proliferation Assays

INS-1 cells were seeded into 96-well plates at a density of 1 × 10^4^ cells/well and cultured overnight. Cells were then treated with varying concentrations of DhHP-6 in media for 24 h. Cell viability was measured using 3-(4,5-dimethylthiazol-2-yl)-2,5-diphenyl-2H-tetrazolium bromide (MTT) assays [47].

### 4.9. Effects of DhHP-6 on H_2_O_2_ Mediated Cell Damage

The oxidative damage model of INS-1 cells was mainly induced by several inducers, such as high glucose [48,49], palmitic acid [50], and hydrogen peroxide [51,52]. In this study, we used H_2_O_2_ as an oxidizing agent, which is commonly used to cause irreversible oxidative damage in various cell models [53]. INS-1 cells were seeded into 96-well plates at a density of 1 × 10^4^ cells/well and cultured overnight. Cells were then treated with 1 mM H_2_O_2_ in the presence of varying concentrations of DhHP-6 for 24 h. Cell viability was measured using MTT assays [52].

### 4.10. Insulin/Lactate Dehydrogenase Secretion Assays

INS-1 cells were seeded into 6-well plates at a density of 4 × 10^5^ cells/well and cultured for 48 h. Cells were washed twice in serum-free media and incubated in Krebs-Ringer bicarbonate HEPES (KRBH) buffer containing 24 mM NaHCO_3_, 1 mM MgCl_2_, 25 mM HEPES, 11.5 mM NaCl, and 5 mM KCl, pH 7.0, supplemented with 1% BSA [54] for 1 h. Cells were then treated with varying concentrations of DhHP-6 in KRBH buffer containing 3.3 or 16.7 mM glucose for 20 min. Supernatants were measured using Rat Insulin ELISA kits.

### 4.11. T2DM Mouse Model and DhHP-6 Treatment

Male C57BL/6J mice (weight, 20 ± 2 g) were induced to produce the T2DM model through high-fat fed (HFD)/STZ feeding as previously described [55]. After adjustable feeding for a week, mice were divided into the normal diet group and HFD (20% protein, 48% carbohydrate, and 22% fat, 20.08 kJ/g) group for four weeks [18]. Fasting blood glucose (FBG) levels of the HFD group receiving injection of STZ (100 mg/kg body weight) ≥11.2 mmol/L were considered diabetic after one week of treatment.

T2DM mice were randomly divided into four groups (*n* = 8) and eight normal mice were taken as the control group. The six groups were treated as follows: (1) normal control group administered physiological saline; (2) DM group administered physiological saline; (3) metformin group of DM mice administered metformin (250 mg/kg body weight); (4) low-dose group of DM mice administered DhHP-6 (20 mg/kg body weight); (5) high-dose group of DM administered DhHP-6 (60 mg/kg body weight). All groups were orally administered equal drug volumes for four weeks.

### 4.12. Measurement of Body Weight and Blood Glucose of T2DM Mice

Body weight and blood glucose levels were measured every week. Prior to measurements, mice were fasted overnight. Blood glucose was determined on a blood glucose meter using test strips (Sino care Inc., Changsha, China) according to the manufacturer’s instructions.

### 4.13. Oral Glucose Tolerance and Insulin Resistance Tests

Mice were fasted overnight prior to OGTT and ITT tests. Blood glucose levels were determined at 0, 30, 60, and 120 min post-glucose (orally administrated 2 g/kg glucose for OGTT) or insulin (intraperitoneally injected 0.75 units/kg for ITT) treatment. Blood glucose was assessed on a blood glucose meter (Sinocare, Changsha, China) and through the use of test strips according to the manufacturer’s instructions. 

### 4.14. Serum Sample Preparation and Determination

At the end of the experimental protocol, blood samples were collected in sterilized centrifuge tubes from the eye socket and centrifuged at 1000× *g* for 15 min to separate the serum. Serum insulin levels were determined using a mouse insulin ELISA kit (Bioss Biotechnology Co., Ltd., Beijing, China). Serum ALT, ALP, and AST activity was determined using commercial kits (Jiancheng Bioengineering Institute, Nanjing, China).

### 4.15. Histochemistry

At the end of the experiment, the liver and pancreas islets were immediately removed after the mice were sacrificed. The left lobe of the liver was stored at −80 °C and the right lobe of the liver was fixed in 4% paraformaldehyde (Solarbio & Technology Co., Ltd., Beijing, China). After dehydration in a gradient of ethanol concentrations, tissues were paraffin embedded and cut into 5 μm thick sections. Slices were stained with hematoxylin and eosin (H&E) dyes, and sections were mounted in neutral deparaffinized xylene medium for microscopic examination using an Olympus BX53 fluorescence microscope (Olympus, Tokyo, Japan).

### 4.16. Statistical Analysis

Data were presented as mean ± standard deviation (SD). Statistical analysis was performed using one-way ANOVA. Differences were considered statistically significant if *p* < 0.05.

## 5. Conclusions

In this study, we found that DhHP-6 had high stability in SGF, SIF, and plasma, and had a higher apparent permeability coefficient in the Caco-2 cell model. Moreover, DhHP-6 was found in the plasma of rats by intragastric administration. DhHP-6 improved oxidative damage induced by H_2_O_2_ and promoted insulin secretion in a concentration-dependent manner in INS-1 cells. In a T2DM mouse model, DhHP-6 can effectively reduce blood glucose and improve the symptoms of insulin resistance. Meanwhile, DhHP-6 restored the structure and function of pancreas islets and improved liver damage. Our study may provide a new strategy for the oral administration of peptides (Figure 10).

## Figures and Tables

**Figure 1 ijms-20-01517-f001:**
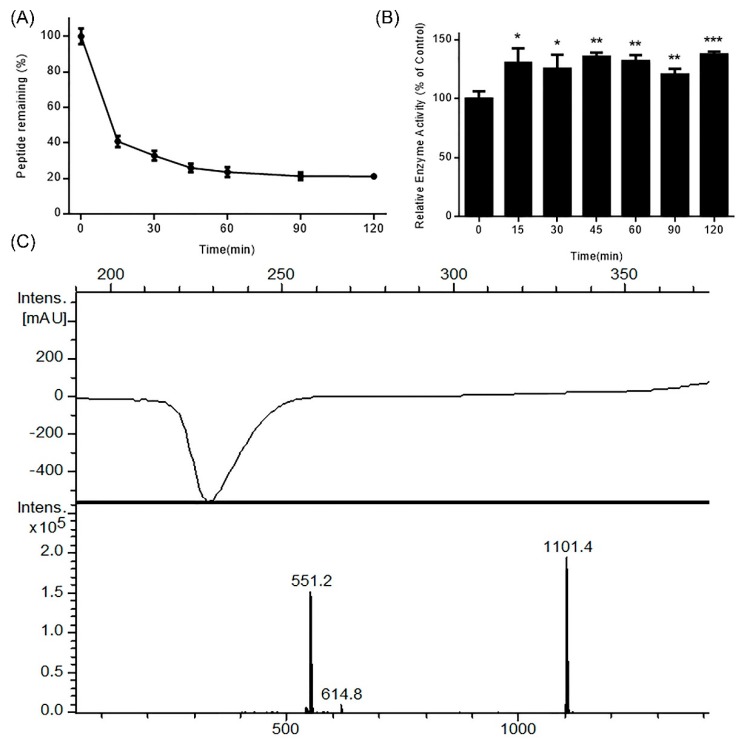
Stability of DhHP-6 in simulated gastric fluid (SGF). (**A**) The remaining DhHP-6 of the degradation products collected at scheduled time was analyzed by HPLC after DhHP-6 was incubated in SGF at 37 °C. (**B**) The degradation products were determined by HPLC-MS. (**C**) The relative enzyme activity of the degradation products was analyzed by the H_2_O_2_–Vitamin C double-substrate method. All results presented are means ± SD from three independent experiments. * *p* < 0.05, ** *p* < 0.01, *** *p* < 0.001 vs. control group.

**Figure 2 ijms-20-01517-f002:**
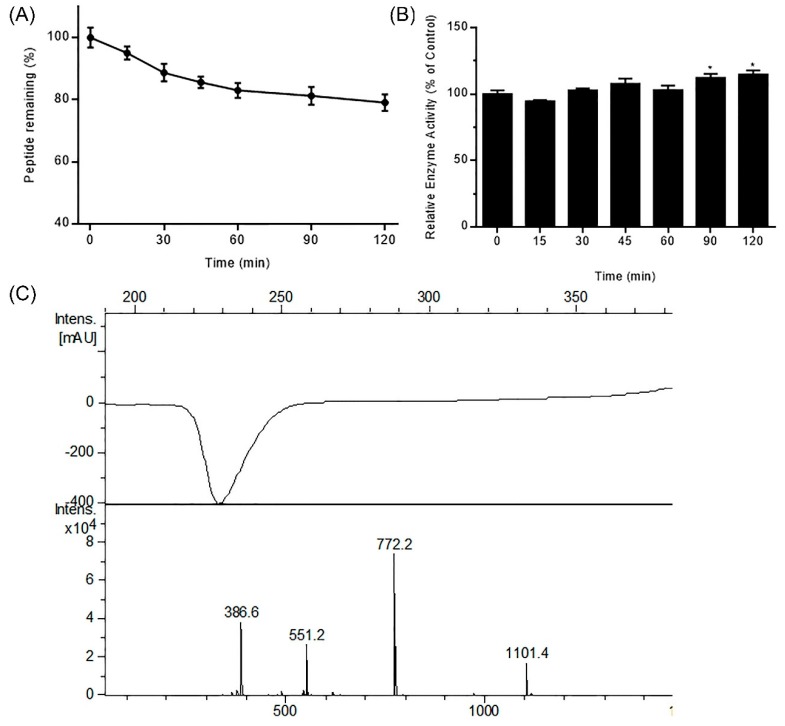
Stability of DhHP-6 in simulated intestinal fluid (SIF). (**A**) The remaining DhHP-6 of the degradation products collected at scheduled time was analyzed by HPLC after DhHP-6 was incubated in SIF at 37 °C. (**B**) The degradation products were determined by HPLC-MS. (**C**) The relative enzyme activity of the degradation products was analyzed by the H_2_O_2_–Vitamin C double-substrate method. All results presented are means ± SD from three independent experiments. * *p* < 0.05 vs. control group.

**Figure 3 ijms-20-01517-f003:**
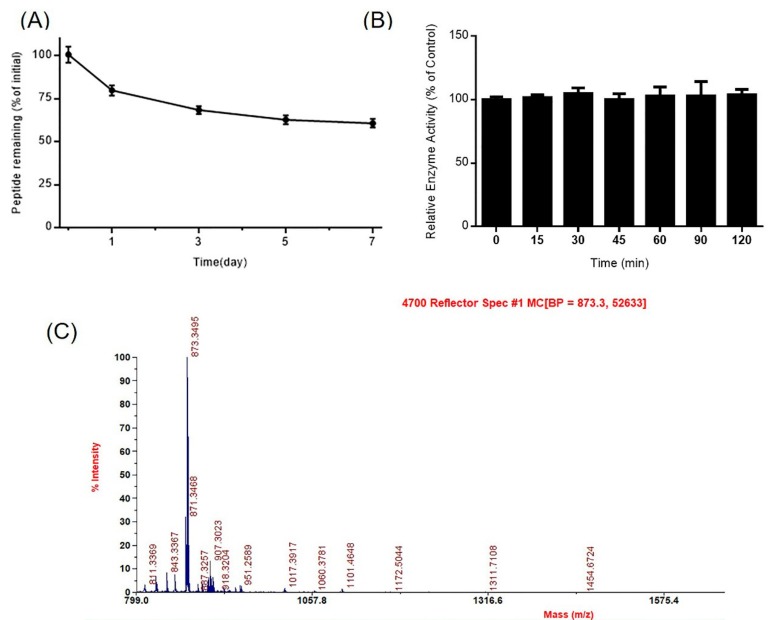
Stability of DhHP-6 in plasma. (**A**) The remaining DhHP-6 of the degradation products collected at scheduled time was analyzed by HPLC after DhHP-6 was incubated in plasma at 37 °C. (**B**) The degradation products were determined by HPLC-MS. (**C**) The relative enzyme activity of the degradation products was analyzed by the H_2_O_2_–Vitamin C double-substrate method. All results presented are means ± SD from three independent experiments.

**Figure 4 ijms-20-01517-f004:**
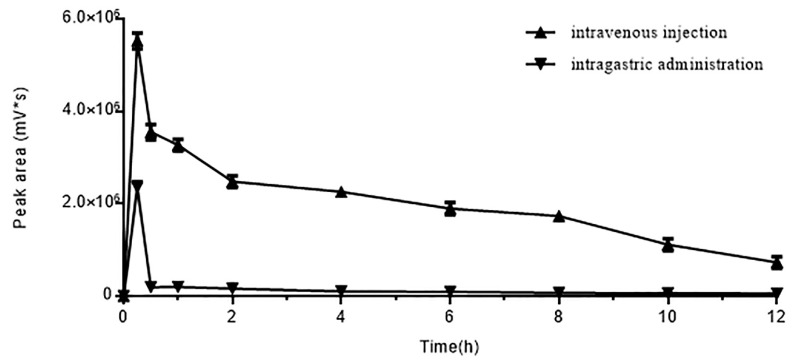
Effect of two administration methods on DhHP-6 absorption. Rats were given the same dose of DhHP-6 by intravenous injection/intragastric administration. The blood was collected at scheduled time and analyzed by HPLC. All results presented are means ± SD, *n* = 5.

**Figure 5 ijms-20-01517-f005:**
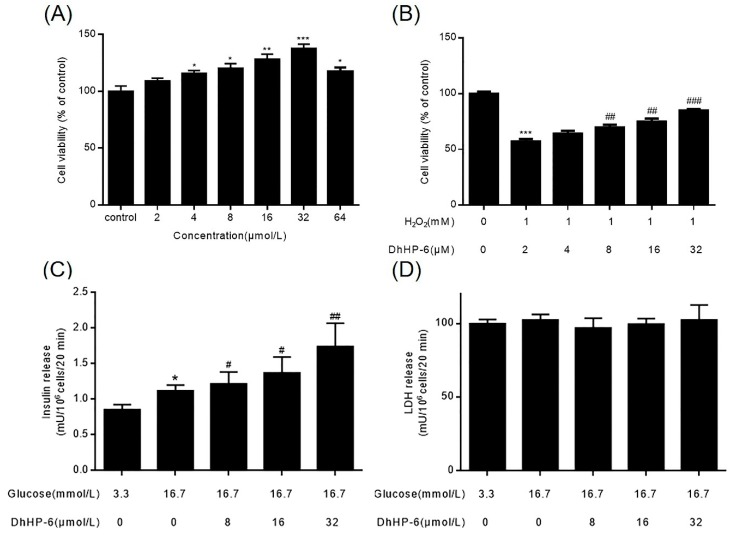
The effect of DhHP-6 on the function and mass of INS-1 cells. (**A**) DhHP-6 promoted the proliferation of INS-1 cells dose-dependently at the concentration less than 64 μmol/L. (**B**) DhHP-6 protected INS-1 cells from the cytotoxicity induced by H_2_O_2_ significantly. (**C**) Glucose (16.7 mmol/L) led to more apparent release of insulin than that at 3.3 mmol/L, but DhHP-6 promoted insulin release of INS-1 cells treated by glucose obviously (16.7 mmol/L). (**D**) DhHP-6 did not stimulate LDH release. All results presented are means ± SD from three independent experiments. * *p* < 0.05, ** *p* < 0.01, *** *p* < 0.001 vs. control group. # *p* < 0.05, ## *p* < 0.01, ### *p* < 0.001 vs. model group.

**Figure 6 ijms-20-01517-f006:**
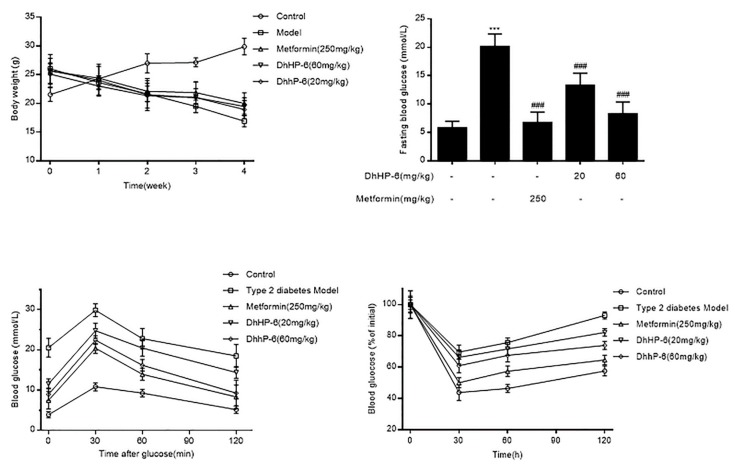
DhHP-6 showed effects on ameliorating the symptoms of type 2 diabetes mellitus (T2DM) mice. (**A**) DhHP-6 had no significant effect on the body weight compared to the model group. (**B**) The blood glucose was tested at the end of the fifth week. (**C**) Glucose tolerance was determined using oral glucose tolerance test (OGTT) assay. (**D**) Insulin resistance was determined using insulin tolerance test (ITT) assay. All results presented are means ± SD, *n* = 8. *** *p* < 0.001 vs. control group. ### *p* < 0.001 vs. model group.

**Figure 7 ijms-20-01517-f007:**
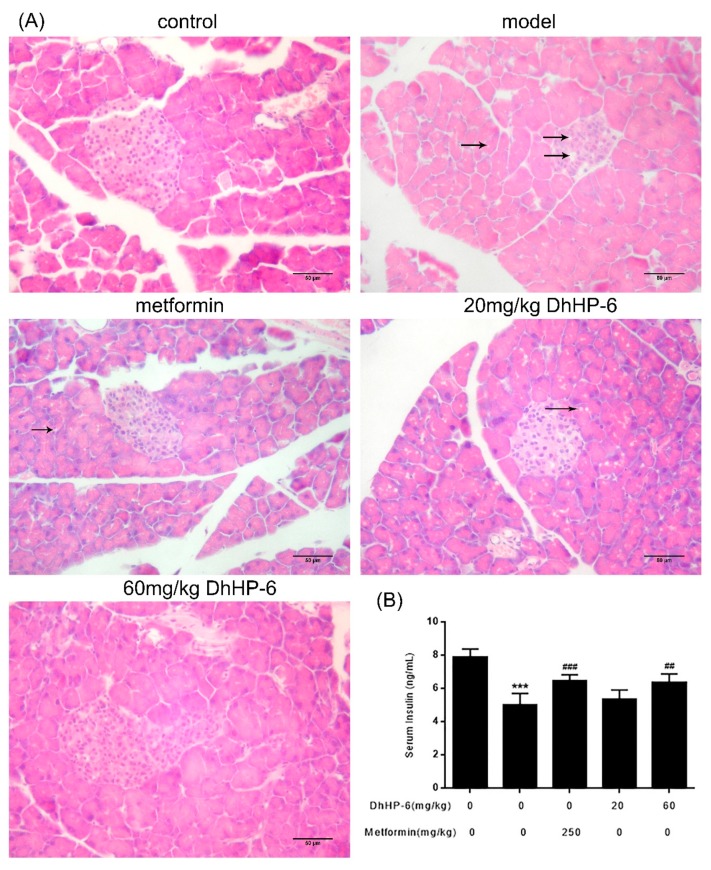
Effect of DhHP-6 on the structure and function of T2DM mice pancreas islets. (**A**) H&E was performed to evaluate the effect of DhHP-6 on the histology of T2DM mice pancreas islets. (**B**) Insulin in serum of T2DM mice was analyzed by ELISA. All results presented are means ± SD, *n* = 8. *** *p* < 0.001 vs. control group. ## *p* < 0.01, ### *p* < 0.001 vs. model group.

**Figure 8 ijms-20-01517-f008:**
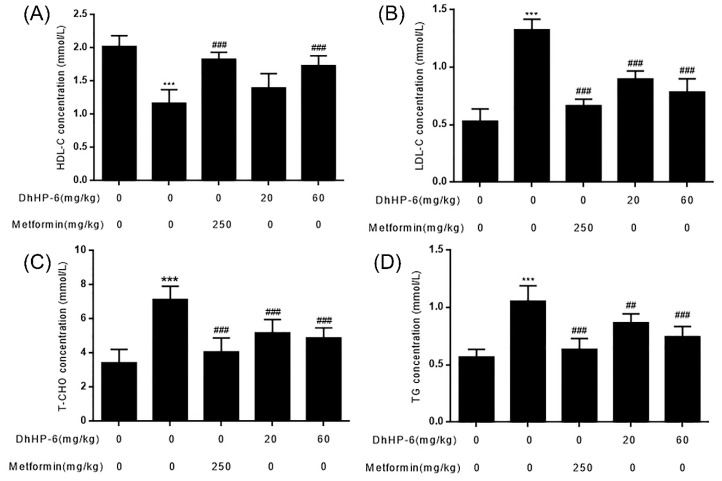
The serum lipid profile was tested on the day of sample collection. (**A**–**D**) HDL-C, LDL-C, T-CHO, and TG in serum were measured. All results presented are means ± SD, *n* = 8. *** *p* < 0.001 vs. control group. ## *p* < 0.01, ### *p* < 0.001 vs. model group.

**Figure 9 ijms-20-01517-f009:**
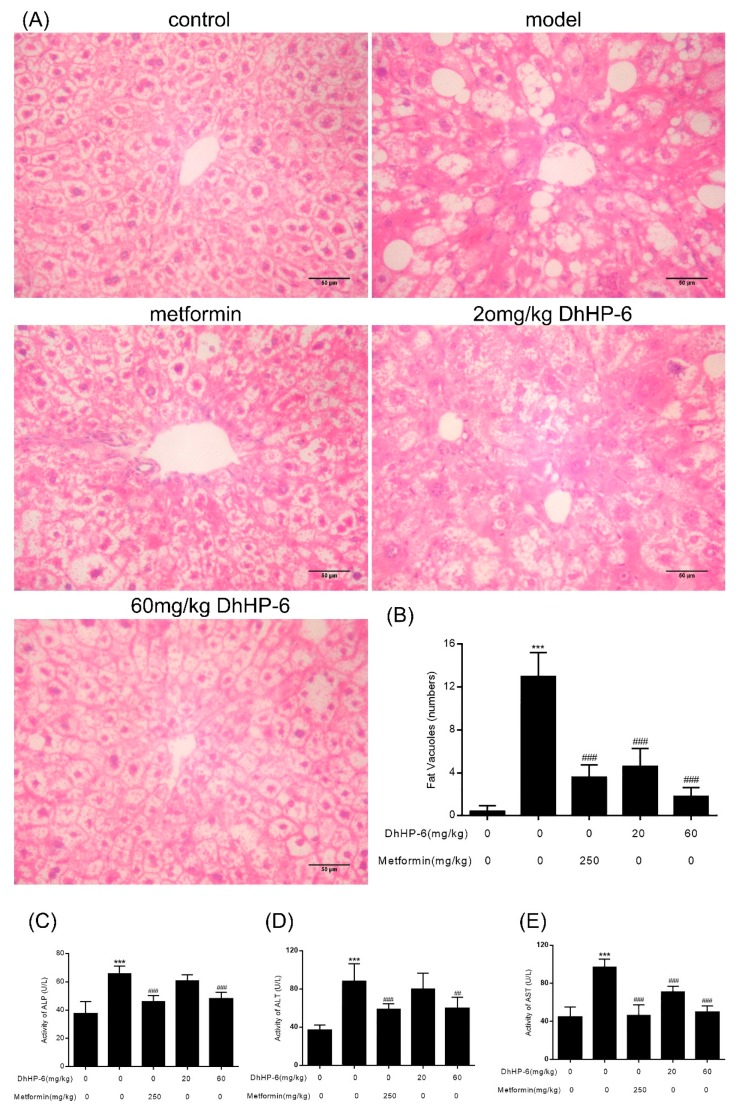
Effects of DhHP-6 on the structure and function of T2DM mice liver. (**A**) H&E was performed to evaluate the effect of DhHP-6 on the histology of T2DM mice liver. (**B**) Histopathological liver damage scores in each group were calculated based on the number of fat vacuoles. (**C–E**) Alkaline phosphatase (ALP), aspartate transaminase (ALT), and alanine transaminase (AST) as liver function markers in serum were measured. All results presented are means ± SD, *n* = 8. *** *p* < 0.001 vs. control group. ## *p* < 0.01, ### *p* < 0.001 vs. model group.

**Figure 10 ijms-20-01517-f010:**
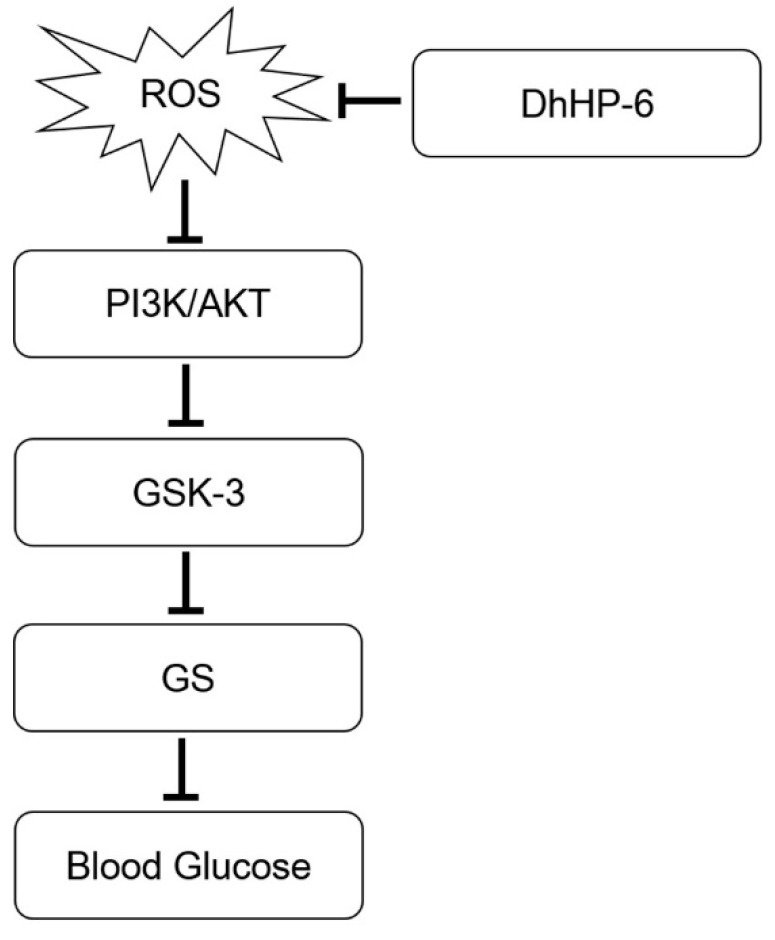
The possible role of DhHP-6 in type 2 diabetes experimental model. ROS impairs PI3K/AKT pathway. AKT stimulation phosphorylates and inactivates GSK-3, which subsequently results in inhibition of GS by phosphorylation. Inactivation of GS blocks glycogen synthesis and increases blood sugar. ROS, Reactive oxygen species; GSK-3, glycogen synthase kinase-3; GS, glycogen synthase.

**Table 1 ijms-20-01517-t001:** Permeability of digestion products in Caco-2 cells. The results are presented as mean ± SD from three independent experiments. * present *p* < 0.05 vs. DhHP-6, ** present *p* < 0.01 vs. DhHP-6.

Compound	Papp AP ⟶ BL (×10^−7^ cm/s)	*p*
DhHP-6	6.08 ± 2.48	0.04275
DhHP-6 in SGF	10.61 ± 3.47 *	0.03959
DhHP-6 in SGF+SIF	18.6 ± 2.95 **	0.00923

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
