# Peer review of "Oral DhHP-6 for the Treatment of Type 2 Diabetes Mellitus"

_ijms, 2019, doi:10.3390/ijms20061517_

Round 1
Reviewer 1 Report
Dear editor,
The manuscript that is authored by Kai Wang et al assessed the utility of DhHP-6 as an oral anti-diabetic drug in-vitro and in-vivo. I suggest accepting the manuscript for publication in the present form.
Merits of this paper:
DhHP-6 is a drug candidate for the treatment of diabetes and diabetes is still a hot topic and there is unmet need for developing new drug candidates.
The manuscript assesses the utility of DhHP-6 as an oral antidiabetic drug in-vitro and in-vivo.
The authors performed the experiments according to the accepted protocols.
Significance of content:
In addition to the in-vivo experiments that give nice and clear results, the authors performed three in-vitro experiments that may help in evaluating the lead-likeness of this drug candidate, 1- stability in simulated gastrointestinal fluid, 2- stability in human plasma, and permeability of digestion products of DhHP-6 in Caco-2 cells.
Quality of presentation:
The authors presented well their work. It is well-written and clear.
Scientific soundness:
This manuscript reports new and important information about DhHP-6. I think, this is important for people from Academia and Industry who is working on developing new drugs for diabetes treatment.
Author Response
Response to Reviewer 1 Comments
Dear Review 1,
Thanks for your letter.
Thanks for your critical examination. Your comments are very helpful for the revision of this manuscript. We are sending the revised manuscript according to the comments. The revised parts are in red and the reply to each point is listed below:
Point 1: Merits of this paper: DhHP-6 is a drug candidate for the treatment of diabetes and diabetes is still a hot topic and there is unmet need for developing new drug candidates.The manuscript assesses the utility of DhHP-6 as an oral antidiabetic drug in-vitro and in-vivo.The authors performed the experiments according to the accepted protocols.
Response 1: Thank you for your comments.
Point 2: Significance of content: In addition to the in-vivo experiments that give nice and clear results, the authors performed three in-vitro experiments that may help in evaluating the lead-likeness of this drug candidate, 1- stability in simulated gastrointestinal fluid, 2- stability in human plasma, and permeability of digestion products of DhHP-6 in Caco-2 cells.
Response 2: Thank you for your comments.
Point 3: Quality of presentation: The authors presented well their work. It is well-written and clear.
Response 3: Thank you for your comments.
Point 4: Scientific soundness: This manuscript reports new and important information about DhHP-6. I think, this is important for people from Academia and Industry who is working on developing new drugs for diabetes treatment.
Response 4: Thank you for your comments.
We appreciate your consideration of our manuscript, and we look forward to receiving comments from the reviewers.
Yours sincerely,
Kai Wang
Reviewer 2 Report
The manuscript “Oral DhHP-6 for the treatment of type 2 diabetes mellitus” discussed about therapeutic effect of DhHP-6 against type 2 diabetes experimental model. In the manuscript, authors have done the both in vitro and in vivo experiments and have provided significant findings to support the hypothesis of the research.
Specific comments and clarification need to be addressed by authors before accepted for publication:
1. In the introduction, need to provide clear justification/problem statement of the studies and also provide epidemiology of type 2 diabetes.
2. The objective of the study need to be re-write in a detailed manner
3. In Figure 4, why authors have used H2O2 induce the oxidative damage? How do you want to interlink the oxidative damage mechanisms and type 2 diabetes? Is there any supportive studies?
4. In the in vivo animal studies, are there any dose dependent studies along with toxicity? Provide the justification for dose fixation
5. Figure 7, need to point out the islets damage in the histological section. Figure 9, I would encourage authors to provide the histological score.
6. The conclusion of the findings are not clear so I would suggest authors to re-write conclusion section and also draw diagram to depicts the role of DhHP-6 in type 2 diabetes experimental model.
Author Response
Response to Reviewer 1 Comments
Dear Review 1,
Thanks for your letter.
Thanks for your critical examination. Your comments are very helpful for the revision of this manuscript. We are sending the revised manuscript according to the comments. The revised parts are in red and the reply to each point is listed below:
Point 1: In the introduction, need to provide clear justification/problem statement of the studies and also provide epidemiology of type 2 diabetes.
Response 1: Thanks for your kindly suggestion. We have provided justification/problem statement of the studies and also provided epidemiology of type 2 diabetes.
Point 2: The objective of the study need to be re-write in a detailed manner.
Response 2: Thanks for your suggestion. We have rewrite the objective of the study.
Point 3: In Figure 4, why authors have used H2O2 induce the oxidative damage? How do you want to interlink the oxidative damage mechanisms and type 2 diabetes? Is there any supportive studies?
Response 3: Thanks for your suggestion. We have provided the reason for using H2O2 induce the oxidative damage. Thanks to your advice, we have also provided some supportive studies to interlink the oxidative damage mechanisms and type 2 diabetes in our manuscript.
Point 4: In the in vivo animal studies, are there any dose dependent studies along with toxicity? Provide the justification for dose fixation.
Response 4: Thanks for your kindly suggestion. Before the study, we conducted an acute toxicity experiment and found that the LD50 of DhHP-6 for male ICR mice by injection was
233.2mg/kg. Regression Equation: Y=4.9987+4.2178X, (α=0.01, r=0.9590),95% confidence interval: 181.4~299.9mg/kg.
Point 5: Figure 7, need to point out the islets damage in the histological section. Figure 9, I would encourage authors to provide the histological score.
Response 5: Thank you for your suggestion. We have pointed out the islets damage in the histological section in figure 7. And we have also provided the histological score in figure 9.
Point 6: The conclusion of the findings are not clear so I would suggest authors to re-write conclusion section and also draw diagram to depicts the role of DhHP-6 in type 2 diabetes experimental model.
Response 6: Thank you for your kindly suggestion. We have rewritten the conclusion section. The mechanism of DhHP-6 in type 2 diabetes remains our further study. We have only drawn a diagram to depict the possible role of DhHP-6 in type 2 diabetes experimental model.
We appreciate your consideration of our manuscript, and we look forward to receiving comments from the reviewers.
Yours sincerely,
Kai Wang
Round 2
Reviewer 2 Report
Thank you for the revised version of manuscript. The authors have addressed all the suggestions and comments adequately, anyhow authors need to address below mentioned comments before accepted for publication.
Comments:
In the methodology section, authors have to re-write the paragraph since authors have provided the results only for histological staining but in the methods section have mentioned Immunohistochemistry so need to be corrected based on the results.
" 4.15 Immunohistochemistry
At the end of the experiment, the liver and pancreatic islets were immediately removed and mice were sacrificed. The left lobe of the liver was stored at -80 °C and the right lobe of the liver was fixed in 4% paraformaldehyde (Solarbio & Technology Co., ltd, Beijing, China). After dehydration in a gradient of ethanol concentrations, tissues were paraffin embedded and cut into 5-μm thick sections. Slices were stained with hematoxylin and eosin (H & E) dyes, and sections were mounted in neutral deparaffinized xylene medium for microscopic examination using an Olympus BX53 fluorescence microscope (Olympus, Japan)."
In the methods section, authors have to include the description, how they calculated histopathological liver damage scores?.
(B) Histopathological liver damage scores in each group. (C, D, E)
Author Response
Dear Review 1,
Thanks for your letter.
Thanks for your critical examination. Your comments are very helpful for the revision of this manuscript. We are sending the revised manuscript according to the comments. The revised parts are in red and the reply to each point is listed below:
Point 1: In the methodology section, authors have to re-write the paragraph since authors have provided the results only for histological staining but in the methods section have mentioned Immunohistochemistry so need to be corrected based on the results.
"4.15 Immunohistochemistry
At the end of the experiment, the liver and pancreatic islets were immediately removed and mice were sacrificed. The left lobe of the liver was stored at -80 °C and the right lobe of the liver was fixed in 4% paraformaldehyde (Solarbio & Technology Co., ltd, Beijing, China). After dehydration in a gradient of ethanol concentrations, tissues were paraffin embedded and cut into 5-μm thick sections. Slices were stained with hematoxylin and eosin (H & E) dyes, and sections were mounted in neutral deparaffinized xylene medium for microscopic examination using an Olympus BX53 fluorescence microscope (Olympus, Japan)."
Response 1: Thanks for your kindly suggestion. We have corrected the title in the methodology section based on the results.
Point 2: In the methods section, authors have to include the description, how they calculated histopathological liver damage scores?
(B) Histopathological liver damage scores in each group. (C, D, E)
Response 2: Thanks for your suggestion. We have included the description for the liver damage scores.
We appreciate your consideration of our manuscript, and we look forward to receiving comments from the reviewers.
Yours sincerely,
Kai Wang
Round 3
Reviewer 2 Report
The revised version of manuscript is acceptable for publication.